# MDM2 Influences ACE2 Stability and SARS-CoV-2 Uptake

**DOI:** 10.3390/v15081763

**Published:** 2023-08-18

**Authors:** Quirin Emslander, Karsten Krey, Sabri Hamad, Susanne Maidl, Lila Oubraham, Joshua Hesse, Alexander Henrici, Katharina Austen, Julia Mergner, Vincent Grass, Andreas Pichlmair

**Affiliations:** 1Institute of Virology, School of Medicine, Technical University of Munich (TUM), 81675 Munich, Germanysabri.hamad@tum.de (S.H.);; 2BayBioMS@MRI—Bavarian Center for Biomolecular Mass Spectrometry at Klinikum Rechts der Isar, Technical University of Munich, 81675 Munich, Germany; 3German Centre for Infection Research (DZIF), Partner site Munich, 81675 Munich, Germany; 4Center of Immunology of Viral Infection (CiViA), Aarhus University, 8000 Aarhus, Denmark

**Keywords:** severe acute respiratory syndrome coronavirus type 2 (SARS-CoV-2), SARS-CoV-2 uptake, MDM2, angiotensin-converting enzyme 2 (ACE2), ubiquitination

## Abstract

Angiotensin-converting enzyme 2 (ACE2) is the central entry receptor for SARS-CoV-2. However, surprisingly little is known about the effects of host regulators on ACE2 localization, expression, and the associated influence on SARS-CoV-2 infection. Here we identify that ACE2 expression levels are regulated by the E3 ligase MDM2 and that MDM2 levels indirectly influence infection with SARS-CoV-2. Genetic depletion of MDM2 elevated ACE2 expression levels, which strongly promoted infection with all SARS-CoV-2 isolates tested. SARS-CoV-2 spike-pseudotyped viruses and the uptake of non-replication-competent virus-like particles showed that MDM2 affects the viral uptake process. MDM2 ubiquitinates Lysine 788 of ACE2 to induce proteasomal degradation, and degradation of this residue led to higher ACE2 expression levels and superior virus particle uptake. Our study illustrates that cellular regulators of ACE2 stability, such as MDM2, play an important role in defining the infection capabilities of SARS-CoV-2.

## 1. Introduction

Angiotensin-converting enzyme 2 (ACE2) is a transmembrane protein regulating the renin–angiotensin–aldosterone system and thereby contributing to blood pressure management [1]. Interestingly, it is also the central entry receptor for several coronaviruses, among them Severe Acute Respiratory Syndrome Coronavirus-2 (SARS-CoV-2) [2], the causative pathogen for the COVID-19 pandemic. Genetic depletion of ACE2 or its blockage with antibodies inhibits SARS-CoV-2 infection in vivo [3], further supporting the central importance of this protein for virus entry. SARS-CoV-2 initiates its uptake by adhering to the receptor binding domain of the glycoprotein spike (S) to ACE2 [2]. To deliver the genome to the cytoplasm, the SARS-CoV-2 particle is either fused at the plasma membrane or in the endosome by employing TMPRSS2 [4] or a cathepsin L-dependent process [5], respectively. These proteases cleave the S protein and enable the fusion of the viral particle with cellular membranes [6]. Several SARS-CoV-2 variants of concern evolved adaptive mutations in the S protein, further underlining the central importance of ACE2. These adaptive mutations led to enhanced binding efficacy of the S/ACE2 interaction site, e.g., the D614G mutation, which stabilized the S-conformation towards an ACE2 binding-competent state [6], as well as T478I and N501Y, which modulated the host range of SARS-CoV-2 [7]. Overall, these amino acid substitutions and others increased the infectivity of the emerging SARS-CoV-2 variants and promoted the global spread of the pathogen [8,9]. 

SARS-CoV-2 predominantly infects the respiratory system. Surprisingly, compared to other organs, ACE2 is not highly expressed in lung cells (https://www.proteinatlas.org/, accessed on 1 May 2023) [10], which gives us a reason to speculate on the relevance of additional proteins promoting entry or alternative receptors that contribute to defining the tropism of the virus. Indeed, it has been reported that the Tyrosine-protein kinase receptor UFO (AXL) can serve as an entry receptor independently of ACE2 [11], and neuropilin-1, which is highly expressed in the bronchus and susceptible brain areas, supports SARS-CoV-2 uptake by bringing virus particles in close proximity to ACE2 [12,13]. Also, matrix metalloproteinases (MMPs) have been shown to promote SARS-CoV-2 entry; e.g., ADAM17 can facilitate the cleavage of the spike, and their inhibition led to the downregulation of SARS-CoV-2 infection [14].

Even though understanding the SARS-CoV-2/ACE2 interface is essential for COVID-19 pathology and therapeutic approaches, surprisingly little is known about the regulation of ACE2 by the host. Particularly, the effect of post-translational modifications on ACE2 and the consequences for SARS-CoV-2 infection are only marginally understood. In a CRISPR/Cas9-mediated screening approach to identify pro-/antiviral host factors, we found that the depletion of MDM2 substantially increased infection with SARS-CoV-2. This E3 ubiquitin ligase is best described for its ability to ubiquitinate p53 and thereby regulate its abundance [15]. Here we show that MDM2 depletion leads to elevated levels of ACE2 on the cell surface, which correlates with superior uptake of SARS-CoV-2 particles. We further show that the mutation of the ubiquitinated Lysine at position 788 into Arginine (K788R) increased the stability of ACE2 and thus its overall abundance, leading to significantly better uptake of SARS-CoV-2 and subsequently higher infection rates.

## 2. Material and Methods

### 2.1. Cell Lines

The following cells were used in the experiments: A549-H2B-mRFP and A549-H2B-mRFP-ACE2 for infection experiments and HEK293R1 for lentivirus production. All cells were cultured as previously described in DMEM (Invitrogen, Carlsbad, CA, USA) with 10% FCS and 1% Penicillin/Streptomycin [16]. All cells were tested to be mycoplasma-free.

Generation of knockout A549-H2B-mRFP-ACE2 cells was performed by cloning multiplexed sgRNA template sequences into the pLentiCRISPRv2 plasmid (Addgene plasmid #52961, Addgene, Watertown, MA, USA). All target sequences were designed with Synthego (Redwood City, CA, USA), except the NTC sequences, which were used as previously published [17] (Appendix A). Lentivirus production, transduction of cells, and antibiotic selection were performed as described previously [18]. In short, lentiviruses encoding puromycin resistance, Cas9, and sgRNAs were added to A549-H2B-mRFP-ACE2, followed by a 7-day puromycin selection (2 µg/mL). All cells were validated for their respective knockouts by sequencing using Synthego Performance Analysis (ICE Analysis, 2019. v3.0. Synthego, Redwood City, CA, USA, 06/2021).

For the generation of A549-H2B-mRFP-ACE2 and A549-H2B-RFP-ACE2 (K788R) cells, the lentiviruses were produced as described before with the pWpi-puro expression vector [16]. To ensure similar lentiviral transduction efficacy, we used a low MOI (0.1) of lentivirus for transduction of A549-H2B-mRFP, followed by antibiotic selection for a stable cell population. Plasmids were sequenced to validate the successful incorporation of the construct and the mutation K788R, respectively.

### 2.2. Virus Stock Preparation

The following viruses were used in this study and were produced as previously described: clinical SARS-CoV-2 WT and SARS-CoV-2-GFP strains were produced by infecting Vero E6 cells (2 days, MOI 0.01) [16] and VSV-S as in [14]. VSV-GFP was produced as in [19], RVFV-GFP was a kind gift from Friedemann Weber, and virus stocks were prepared as previously published [20].

### 2.3. Virus Infection and Live-Cell Imaging Experiment

A549-H2B-mRFP-ACE2, A549-H2B-mRFP-ACE2 (K788R), and the knockout cell line were seeded one day prior to the experiment in 96-well plates. Cells were infected with the reporter viruses SARS-CoV-2-GFP (MOI: 3), YFV-GFP (MOI: 1), VSV-GFP (MOI: 0.01), RVFV-Katuschka (MOI: 0.1) and HSV-1-GFP (MOI: 0.1). The infection was measured with the IncuCyte S3 Live-Cell Analysis System (Essen Bioscience) by acquiring the GFP, RFP, and phase contrast signal over 24–72 h post-infection, with 3–4 h intervals depending on the kinetics of the virus using a 4x objective. The raw data were processed with the internal IncuCyte S3 Software (Essen Bioscience; version 2020C rev1) by normalizing the integrated intensity of the GFP (virus) signal to either the integrated intensity of the RFP signal (cell) or the confluency determined by the phase contrast channel (cell). For the calculation of significance, a two-sample *t*-test *p* < 0.05 (*), <0.01 (**), <0.001 (***).

We then chose, for each infection, the maximum signal value of each knockout and calculated its percentage change compared to the NTC:(1)$$\%\;change=\frac{mean(knockout)}{mean(NTC)}-1$$

Life-cell imaging experiments were conducted as explained above, but cells were pre-treated for 4 h with MG132 (medchem, HY-13259) or PR-619 (medchem, HY-11814). The inhibitors were solubilized in DMSO and, for the experiments, diluted in cell culture medium. Cells were either not infected to test the cytotoxicity or pre-treated with the drugs for 4 h and then infected with the reporter virus SARS-CoV-2-GFP (MOI: 3) to monitor the effect of the drug on the virus.

### 2.4. Full Proteome MS Sample Preparation

Each of the replicates of A549-H2B-mRFP-ACE2 cells (MDM2 knockouts and NTCs) was infected with SARS-CoV-2 (MOI 3). The samples were harvested in SDC buffer (4% SDC, 100 mM Tris-HCl, pH 8.5), heat-inactivated (95 °C, 10 min), and sonicated (4 °C, 15 min, 30 s on/30 s off, high settings). In short, protein concentrations were measured by the BCA assay (Pierce, Appleton, WI, USA) according to the manufacturer’s instructions. 50 µg of protein material were reduced and alkylated for 5 min at 45 °C with TCEP (10 mM) and CAA (40 mM). For each sample, 50 μg of protein material was digested overnight at 37 °C using trypsin (1:100 *w*/*w*, enzyme/protein, Promega, Madison, WI, USA) and LysC (1:100 *w*/*w*, enzyme/protein, Wako, Hiroshima-shi, Japan).

For proteome analysis, 20 μg of peptide material was desalted using SDB-RPS StageTips (Empore, Stellarton, NS, Canada). Samples were diluted with 1% trifluoroacetic acid (TFA) in isopropanol to a final volume of 200 μL and loaded onto StageTips, subsequently washed with 1% TFA in isopropanol and 0.2% TFA/5% acetonitrile (ACN, Concord, CA, USA). Peptides were eluted with 1.25% ammonium hydroxide (NH_4_OH) in 60% ACN and dried using a SpeedVac centrifuge (Eppendorf, Concentrator Plus, Hamburg, Germany). They were resuspended in 0.1% FA prior to LC–MS/MS analysis. Peptide concentrations were measured optically at 280 nm (Nanodrop 2000, Thermo Scientific, Waltham, MA, USA) and subsequently equalized using 0.1% FA.

### 2.5. LC-MS/MS Data Acquisition

Full proteome samples were measured on an Eclipse mass spectrometer (Thermo Fisher Scientific, Waltham, MA, USA) coupled on-line to a Dionex Ultimate 3000 RSLCnano system (Thermo Fisher Scientific). Peptides were reconstituted in 0.1% FA and delivered to a trap column (ReproSil-pur C18-AQ, 5 μm; Dr. Maisch, 20 mm × 75 μm, self-packed) at a flow rate of 5 μL/min in 100% solvent A (0.1% FA in HPLC-grade water). After 10 min of loading, peptides were transferred to an analytical column (ReproSil Gold C18-AQ, 3 μm; Dr. Maisch, 400 mm × 75 μm, self-packed) at 50 °C and separated using an 80-min linear gradient from 4% to 32% of solvent B (0.1% FA, 5% DMSO in ACN) at a 300 nL/min flow rate.

The mass spectrometer was operated in data-dependent acquisition and positive ionization modes. MS1 full scans (360–1300 m/z) were acquired in the orbitrap with a resolution of 60,000, a normalized automatic gain control (AGC) value of 100%, and a maximum injection time of 50 ms. Peptide precursor selection for fragmentation was carried out using a fixed cycle time of 2 s. Only precursors with charge states from 2 to 6 were selected, and dynamic exclusion of 35 s was enabled. Peptide fragmentation was performed using higher-energy collision-induced dissociation (HCD) and a normalized collision energy of 30%. The precursor isolation window width of the quadrupole was set to 1.3 m/z. MS2 spectra were acquired in the orbitrap with a resolution of 15,000, a fixed first mass of 100 m/z, a normalized AGC target value of 200%, and a maximum injection time of 22 ms.

### 2.6. MS Data Processing and Analysis

Raw MS data files of the experiments conducted in DDA mode were processed with MaxQuant (version 2.0.3.1.) using the default settings and label-free quantification (LFQ) (LFQ min ratio count 2, normalization type classic) and intensity-Based Absolute Quantification (iBAQ) enabled. Spectra were searched against forward and reverse sequences of the reviewed human proteome, including isoforms (Uniprot, UP000005640), by the built-in Andromeda search engine. 

The protein groups were further analyzed using Perseus (Version: 1.6.15.0). The LFQ values of 4941 proteins were imported from the proteinGroups.txt file, and the columns were filtered for the default settings of reverse, only identified by site and contaminants. The LFQ values were log2 transformed, and the protein groups were filtered for at least 2 valid values per grouping. The missing values were replaced by the normal distribution (1.8 downshift, 0.3 width, separately for each column). The two conditions, now including 3767 protein IDs, were analyzed with a two-sample *t*-test, leading to 267 significant changing proteins (two-sided, permutation-based FDR, 0.05 FDR, 0.1 S0, 250 randomizations) (Appendix A).

### 2.7. Upstream Regulator Analysis

For the identification of global upstream regulators, significant hits from the full proteome analysis were processed in the Ingenuity pathway analysis software (version 84978992, Qiagen, Hilden, GER). The core analysis was performed using the default settings, including the Ingenuity Knowledge Base as the reference set for *p*-value calculations as well as “direct and indirect relationships” for upstream regulator analysis. After analysis, only upstream regulators were considered significant, exceeding a *p*-value < 0.05.

### 2.8. VSV-S and VSV Infection

To assess whether the observed effect on SARS-CoV-2 was uptake-dependent, we used a VSV pseudovirus that displayed the SARS-CoV-2 spike (S) protein on its surface (VSV-S) and a parental VSV as a control. The day prior to infection, 8e3 A549-H2B-mRFP-ACE2 MDM2 knockout and A549-H2B-mRFP-ACE2 NTC or A549-H2B-mRFP, A549-H2B-mRFP-ACE2, and A549-H2B-mRFP-ACE2 (K788R) cells were seeded in a 96-well plate. The cells were infected with VSV-S (MOI: 3) and VSV-GFP (MOI: 0.01). The infection was measured with the IncuCyte S3 Live-Cell Analysis System (Essen Bioscience) as above. The raw data were processed with the internal IncuCyte S3 Software (Essen Bioscience; version 2020C rev1) by normalizing the integrated intensity of the GFP (virus) signal to either the integrated intensity of the RFP signal (cell) or phase (cell). The resulting normalized signal was then processed by median filtering and then by a Savitzky–Golay filter. Significance was determined by comparison of each of the two conditions for VSV and VSV-S to the control using a two-sample *t*-test for each time point (with a *p*-value ns > 0.05, * < 0.05, ** < 0.01, *** < 0.01).

### 2.9. Virus-like Particle Uptake Assay and Immunofluorescence Staining

For the generation of SARS-CoV-2 VLPs, HEK293T cells were transiently transfected with plasmids encoding S, M, N, E, and CD63~BlaM and harvested from conditioned medium 72 h post-transfection as previously described [21]. Then, 1.6 × 10^4^ A549-H2B-mRFP-ACE2 and A549-H2B-mRFP-ACE2 (K788R) cells were seeded in a 96-well plate one day prior. Cells were washed with PBS (4 °C), and 100 µL of DMEM (10% FCS and 1% Penicillin/Streptomycin) plus 100 µL of VLPs were added per well. Additionally, CellTracker™ Orange CMRA Dye (Thermo Fisher Scientific, Waltham, MA, USA, C34551) was added to the medium/VLP mix according to the manufacturer’s instructions. The plate was incubated for 30 min on ice to enable simultaneous uptake and then moved to the incubator (37 °C, 5% CO_2_) for 1 h. The medium was aspirated, and the cells were washed three times with PBS, fixed with 4% PFA for 15 min at RT, and washed three times with PBS. The cells were permeabilized with 0.1% Triton-X100 at RT, washed with PBS, blocked with 5% FCS (1 h, RT), and washed with PBS. The VLPs were stained with a previously published in-house antibody dilution (1:1000) in 5% FCS for 1 h at RT in the dark, followed by PBS washing. 

For staining of HA-tagged ACE2 levels, cells were processed as above. The HA-tag was stained with (1:500) an antibody dilution (Cell Signaling, Danvers, MA, USA, 3724S) in 5% FCS for 1 h at RT in the dark. 

Super-resolution Images were acquired using a Zeiss LSM 900 (Zeiss, Oberkochen, Germany) with an Airyscan 2 detector system using a Plan-APOCHROMAT 63×/1.4 Oil DIC Objective. Post-acquisition linear adjustments of image brightness were made using the ImageJ software version 1.53. Confocal images were acquired using an Olympus FV10i microscope (Olympus, Tokyo, Japan) with a 60×/1.2 water immersion objective.

### 2.10. qRT-PCR Analysis

The knockout, A549-H2B-mRFP-ACE2 and A549-H2B-mRFP-ACE2 (K788R) cells were seeded (130.000 cells) one day prior to the experiment. For SARS-CoV-2 transcript quantification, cells were infected with SARS-CoV-2 (MOI: 3) and incubated for 1 h, 12 h or 24 h post-infection. To quantify ACE2 levels, cells were not infected. The RNA was isolated using the NucleoSpin RNA Plus kit (Macherey-Nagel, Duren, Germany) according to the manufacturer’s protocol. Total RNA was used for reverse transcription with PrimeScript RT with a gDNA eraser (Takara, Kusatsu, Japan). For relative transcript quantification, PowerUp SYBR Green (Applied Biosystems, Waltham, MA, USA) was used. The previously published primers 5′-TTACAAACATTGGCCGCAAA-3′ (sense) and 5′-GCGCGACATTCCGAAGAA-3′ (antisense) were used for the SARS-CoV-2 nucleocapsid gene quantification [22] and 5′-CATTGGAGCAAGTGTTGGATCTT-3′ (sense) and 5′-GAGCTAATGCATGCCATTCTCA-3′ (antisense) for ACE2 quantification [23]. For normalization, the previously published primers 5′-GATTCCACCCATGGCAAATTC-3′ (sense) and 5′-GGATCTGCTGCATCTGCTTG-3′ (antisense) for the GAPDH gene [24] or housekeeping gene for the RPLP0 gene were used as previously described [14]. Data were analyzed in GraphPad Prism (V.9). RNA level changes were analyzed by one-way ANOVA, comparing the 24 h levels of the knockouts to the 24 h values of the NTC, with *p* > 0.12 (ns), <0.033 (*), <0.002 (**), <0.001 (***).

### 2.11. Western Blotting

For western blot analysis, A549-H2B-mRFP-ACE2 and A549-H2B-mRFP-ACE2 (K788R) cells were seeded for one day in a 6-well format. Cells were washed with PBS and lysed in 200 µL of SSB buffer (62.5 mM Tris-HCl pH 6.8, 2% SDS, 10% Glycerol, 50 mM DTT, 0.01% Bromphenol blue), followed by boiling at 95 °C for 5 min. The protein concentration of the samples was measured using Pierce 660 nm Protein Assay Reagent with the addition of an ionic detergent compatibility reagent (Thermo Fisher Scientific). 5 µg cell lysate was resolved on a NuPAGE 1 mm 4–12% Bis-Tris gel (Invitrogen) in 1× NuPAGE MES SDS Running Buffer (Invitrogen). For the PageRuler Prestained Protein Ladder (Thermo Fisher Scientific), 5 µL was used as a marker. The separated proteins were subsequently transferred to a 0.45 µm nitrocellulose membrane (BioRad, Hercules, CA, USA) at 100 V for 1 h in pre-cooled wet-blotting transfer buffer (25 mM Trizma base, 0.192 M Glycine, pH 8.3, 20% methanol). The membrane was blocked in 5% skim milk in PBS-T buffer (phosphate buffered saline with the addition of 0.25% Tween-20) for 1 h at room temperature under constant agitation and rinsed twice with PBS-T before being incubated with antibodies. All antibodies were diluted with 5% milk in PBS-T. The protein of interest was detected using the primary antibody against ACE2 (abcam, Waltham, MA, USA, ab272690, 1:1000 dilution) and the secondary antibody coupled to horseradish peroxidase rb-HRP (Agilent Dako, Glostrup, Denmark, P0448, 1:2500 dilution). The membrane was washed 3× for 5 min with PBS-T in between and after incubation with primary and secondary antibodies. Before reprobing the membrane with the ACTB-HRP (Santa Cruz, Dallas, TX, USA, sc-47778, 1:2500 dilution) antibody, bound antibodies were stripped using Restore Western Blot Stripping Buffer (Thermo Fisher Scientific), and the membrane was washed 3× for 5 min with PBS-T. The membrane was incubated with SuperSignal West Femto Maximum Sensitivity Substrate (Thermo Fisher Scientific) prior to developing the membrane using the ChemiDoc XRS+ Imager. Optical density was determined with ImageJ (1.52 k, 1.8.0_172), and statistical analysis was performed with GraphPad Prism (V.9), performing a *t*-test.

## 3. Results

### 3.1. Selection of Knockout Candidates Based on Multi-Omics Profiling Data

We selected 21 host proteins for a CRISPR/Cas9-based knockout screen targeting different pathways and contributing to various cellular functions that may be important for SARS-CoV-2 replication or spread. These proteins were selected based on multi-omics profiling of SARS-CoV-2 infection [16], in which they were significantly affected in their mRNA or protein expression or were identified as interaction partners of SARS-CoV-2 proteins. We particularly focused on interactions between ORF7B and ORF3, two multifunctional SARS-CoV-2 proteins necessary for SARS-CoV-2 spread. ORF7B localizes in the Golgi and the virion [25]. It is also reported to induce several cytokines and growth factors and mediate apoptosis regulation [26]. ORF3a is an accessory protein that blocks autolysosome fusion [27]. It is frequently mutating and induces different innate immune responses [28]. ORF7B and ORF3a both interact with the syntaxins STX6, STX10, STX12, and STX16, necessary for various roles in vesicle trafficking [29], the receptor-type tyrosine phosphatase PTPRM, the transmembrane protein DCBLD2, and the solute carrier SLC30A1. The endosomal transport protein VTI1A [30], the SHC transforming protein, and the G-protein coupled receptor 39 (GPR39) interacted with ORF7b, while the cell adhesion proteins PCDHAC2, PCDHGC3, NECTIN3, and CDH16 mainly interacted with ORF3. Matrix metalloproteinase MMP15 interacted with ORF7b and ORF3 and was significantly upregulated in the transcriptome after infection. The TGFB1 regulator, LTBP1 [31], interacted with ORF8. The cell surface protein TNS4 was highly upregulated in the full proteome after the SARS-CoV-2 infection. The transcript levels of OSMR were highly upregulated by SARS-CoV-2. Also, the transcripts of the E3-ubiquitin ligase MDM2 were significantly upregulated by the SARS-CoV-2 infection. The latter was particularly interesting because several publications suggested drugging this E3 ligase to increase p53 levels to counteract SARS-CoV-2 infection [32,33].

### 3.2. MDM2 Knockout Specifically Enhances SARS-CoV-2 Infection

We used a CRISPR/Cas9-based knockout strategy to deplete the described candidates in human lung carcinoma epithelial A549 cells expressing the SARS-CoV-2 entry receptor ACE2 and H2B-mRFP (A549-H2B-mRFP-ACE2) (Appendix A). The knockout efficiency was confirmed by sequencing. As a positive control for our screen, we knocked out lymphocyte antigen 6e (LY6E), a protein that impairs efficient entry for human Coronaviruses [34]. We used a recombinant SARS-CoV-2 that expressed green fluorescent protein (GFP) upon infection (SARS-CoV-2-GFP) and performed continuous live-cell imaging to monitor the effect of the selected candidate on viral spread. We normalized the integrated GFP expression values to the integrated mRFP signal (representative of the cell count) and plotted the maximum value of each infection normalized to the non-targeting control (NTC) (Figure 1A). Our screening approach revealed several effects on SARS-CoV-2 replication. As expected, LY6E depletion led to a significantly increased accumulation of GFP, confirming the antiviral activity of this protein and the validity of the assay. Notably, three tested candidates, MDM2, SLC30A1, and GPR39, displayed a substantial increase in GFP signal (Figure 1B). Interestingly, the strongest decrease in infection was detected for the knockouts of three vesicular trafficking proteins (VTI1A, STX12 and STX16), highlighting the important role of vesicular trafficking for SARS-CoV-2 infection and replication. To test whether the identified proteins were specifically important for SARS-CoV-2, we complemented the analysis of SARS-CoV-2 with growth analyses of unrelated viruses, namely Vesicular Stomatitis Virus (VSV) and Yellow Fever Virus (YFV). As expected, the knockout of the restriction factor LY6E did not affect VSV and YFV but showed exclusive antiviral activity for SARS-CoV-2 (Figure 1B–D). Interestingly, the antiviral effect of syntaxin knockouts appeared to be reversed for VSV, particularly for STX16. We focused on the top four pro-viral and top three antiviral knockouts in the SARS-CoV-2 screen (including all the syntaxins) and the top pro- and antiviral hits of YFV and VSV. These candidates were further tested for their effect against the Rift Valley Fever Virus (RVFV) and Herpes Simplex Virus (HSV-1). Depletion of GPR39, a regulator of heat shock proteins and the hedgehog pathway [35], showed highly virus-specific effects: While GRP39 depletion increased SARS-CoV-2 replication, it decreased the propagation of all other viruses tested (Figure 1B–F). Better growth of SARS-CoV-2 and VSV was seen upon knockout of SLC30A1 (Figure 1B,C), which has been reported to elevate intracellular zinc levels and hence inhibit caspase activation and apoptosis [36,37]. Depletion of SHC1 increased the growth of VSV, YFV HSV-1, and RVFV but slightly inhibited the growth of SARS-CoV-2 (Figure 1B–F). The knockout of NECTIN3 was strongly antiviral for YFV, slightly repressive for VSV, and moderately pro-viral for SARS-CoV-2. No significant changes were observed for RVFV and HSV-1 (Figure 1B–F). However, among all candidates tested, depletion of MDM2 showed the strongest pro-viral effect and was exclusively important for SARS-CoV-2 (Figure 1A–F and Appendix A). This phenotype was very reminiscent of LY6E knockout, which resulted in an infection kinetic of SARS-CoV-2 that was comparable to MDM2 deletion, i.e., an early onset of the infection signal and a rapid increase of the fluorescent signal (Appendix A).

### 3.3. MDM2 Knockout Stabilizes ACE2 and Increases SARS-CoV-2 Replication

We next confirmed the obtained results by quantifying SARS-CoV-2 nucleocapsid RNA between 12 and 24 h post-infection in our knockout cells. Considering the accumulation of viral RNA in NTC cells over time, only three knockout cell lines led to a significantly higher increase in SARS-CoV-2 levels. Notably, in line with our screening approach, the highest increase could be seen in MDM2 knockout cells (Figure 2A and Appendix A). In contrast, among all tested knockouts, depletion of STX16 most strongly reduced viral accumulation. Thus, we could validate the screening results and confirm that depletion of MDM2 led to a significant increase in SARS-CoV-2 replication, while STX16 had the opposite effect.

Given the notable effect of MDM2 knockout, we further focused on the antiviral activity of this protein in the context of the SARS-CoV-2 infection. The comparable infection kinetics and specificity in MDM2 and LY6E-depleted cells led us to hypothesize that MDM2 may impair SARS-CoV-2 uptake, as reported for LY6E [34]. To test this, we infected the MDM2-depleted and control cells with a pseudotyped VSV-GFP reporter virus bearing the SARS-CoV-2 spike protein (VSV-S). We observed a significantly higher GFP signal in MDM2 knockout cells as compared to NTC cells (Figure 2B). As an additional control, we infected MDM2 knockout and NTC cells with the parental VSV-GFP but did not observe any significant difference in the GFP signal (Figure 2C). Collectively, these data showed that depletion of MDM2 specifically increased SARS-CoV-2 uptake but did not affect the uptake of other tested viruses. Moreover, it showed that MDM2 serves as a prominent restricting factor for SARS-CoV-2 entry.

Since MDM2 is an E3 ubiquitin ligase mediating proteasomal degradation of substrates, we proceeded to use proteomics analysis employing liquid chromatography coupled to mass spectrometry (LC–MS/MS) to analyze the global changes in the proteome after depletion of MDM2. In total, we could detect expression values for 4941 proteins in all tested conditions, of which 3767 were used for analysis. Among them, 267 proteins showed significant differences in their expression levels in MDM2 knockout cells as compared to NTC cells (Figure 2C, Appendix A). We performed an upstream regulator analysis on these hits to identify functional clusters and their responsible inducers that could explain the SARS-CoV-2-specific effect (Appendix A). As expected, this analysis identified p53 as an activated regulator (57 of the 267 proteins were regulated by p53) that was specifically enriched in the absence of MDM2 (*p*-value < 2.14 × 10^−14^). Surprisingly, a direct comparison between MDM2 knockout and control cells revealed a significantly increased protein abundance of ACE2 in the absence of MDM2 (Figure 2D). Therefore, the knockout of MDM2 could increase the expression levels of ACE2, plausibly explaining enhanced infection and selective uptake of SARS-CoV-2. Indeed, ACE2 immunostaining showed that MDM2 knockout cells displayed higher levels of ACE2, including at the cell surface, as compared to NTC cells (Appendix A). Collectively, these data suggested that MDM2 controls ACE2 expression levels and that MDM2 depletion led to increased SARS-CoV-2 uptake, likely due to increased levels of ACE2.

### 3.4. Lysine 788 within ACE2 Mediates ACE2 Stability and SARS-CoV-2 Replication

In search of a potential molecular mechanism for the enhanced infectivity of MDM2 knockout cells, we mined a database for post-translational modifications [38] and identified ACE2 Lysine K788 as the only ubiquitination site accessible for cytosolic MDM2 (Appendix A). Importantly, this ACE2-K788 site has already been identified as a target of the E3-ligase MDM2 in patients suffering from idiopathic pulmonary arterial hypertension. It has been shown that MDM2 is upregulated in these patients and that MDM2 ubiquitinates ACE2 at position K788, which leads to a subsequent downregulation of ACE2 through proteasomal degradation. Also, mutation of this site or inhibiting ACE2-expressing cells with a proteasome inhibitor can protect ACE2 from proteasomal degradation, commonly induced by K48-ubiquitination [39]. However, while the consequences of this site for SARS-CoV-2 infection have not been investigated so far, the regulation of ACE2 levels by MDM2-dependent ubiquitination is in line with our infection experiments.

To confirm that MDM2 may target K788 for ubiquitination and subsequent degradation of ACE2, we mutated Lysine 788 into Arginine (K788R), following the hypothesis that preventing MDM2-dependent ubiquitination would stabilize ACE2 and lead to overall higher infection rates. We used lentiviral gene transfer to trans-complement ACE2-negative A549-H2B-mRFP cells with wild-type ACE2 (A549-H2B-mRFP-ACE2) or with a K788R mutant (A549-H2B-RFP-ACE2 (K788R)). Notably, the expression level of the ACE2 (K788R) mutant protein was clearly increased as compared to the expression levels of cells expressing wt ACE2, despite comparable levels of the β-actin (β-ACT) control by 60% (Figure 3A and Appendix A). Importantly, evaluation of the Ace2 transcript levels by quantitative PCR indicated identical Ace2 mRNA expression in both tested cell lines (Figure 3B), supporting that the increase of ACE2-K788R on the protein level was due to increased stability of the mutant rather than higher mRNA expression. To test if the increase in ACE2 level leads to better infection, we infected these cell lines with similar amounts of SARS-CoV-2-GFP reporter virus and monitored the accumulation of the fluorescent signal over time. Indeed, we observed significantly increased infection in the ACE2 (K788R) cells as compared to ACE2-wt controls (Figure 3C). Moreover, further analysis of the obtained pictures showed that more cells got infected with SARS-CoV-2-GFP in ACE2 (K788R) cells as compared to wt ACE2-expressing cells (Appendix A). To test if degradation via the proteasome influences ACE2 levels and thereby SARS-CoV-2 uptake, we treated A549-ACE2 cells with the proteasome inhibitor (MG132) and monitored SARS-CoV2 replication. However, proteasomal inhibition impaired SARS-CoV-2 replication, indicating that proteasomal activity is in general required for virus replication (Appendix A), which is in line with previous findings [16,40]. We speculated that inhibition of deubiquitinating enzymes (DUBs) would increase ACE2 degradation and thus reduce the infectivity of SARS-CoV-2. However, the broadly active deubiquitinase inhibitor (PR-619) did not considerably affect SARS-CoV-2 replication at non-cytotoxic concentrations (0.5 µM) (Appendix A). More specific inhibitors and further functional experiments would be required to identify a chemical compound that targets the described pathway to reduce cellular ACE2 levels.

### 3.5. ACE2-K788R Increases SARS-CoV-2 Particle Uptake

These experiments prompted us to test whether ACE2 (K788R) may allow for better viral uptake. To test this, we infected A549-H2B-mRFP, A549-H2B-mRFP-ACE2, and A549-H2B-mRFP-ACE2 (K788R) cells with VSV-S and VSV GFP reporter viruses, respectively, and monitored GFP expression over time (Figure 4B,C). As expected, A549-H2B-RFP cells that were not trans-complemented with ACE2 could not be infected with VSV-S, while expression of ACE2 led to GFP accumulation over time. Notably, the number of cells infected with VSV-S was significantly higher in A549-H2B-mRFP-ACE2 (K788R) cells as compared to A549-H2B-mRFP-ACE2, further indicating that the uptake of VSV-S was superior in the ACE2 (K788R) cell line (Appendix A). Importantly, the infection with VSV was identical in all three tested cell lines (Figure 4C). Moreover, we infected A549-H2B-RFP-ACE2 and A549-H2B-mRFP-ACE2 (K788R) with SARS-CoV-2 and detected significantly higher SARS-CoV-2 nucleocapsid RNA levels at 1 h post-infection in A549-H2B-RFP-ACE2-K788R cells as compared to wt ACE2-expressing cells (Figure 4A).

To underline the better binding of SARS-CoV-2 to ACE2 (K788R) over ACE2-wt expressing cells, we compared the adhesion of SARS-CoV-2 virus-like particles (VLP) [21] on the two different cell lines using super-resolution microscopy. We expected to observe a higher amount of VLPs on the surface of A549-H2B-mRFP-ACE2 (K788R) as compared to A549-H2B-RFP-ACE2 cells. To test this, we exposed both cell lines to identical amounts of VLPs on ice, followed by a temperature shift to 37 °C for 1 h. In line with the notion that ACE2-K788R increased uptake efficiency, we could observe an increased association of VLPs to the cell surface, as evaluated by staining for Spike (Figure 4D). Taken together, these experiments highlight the specific effect of MDM2-driven ACE2 regulation on SARS-CoV-2 entry.

## 4. Discussion

Here we selected host factors identified from SARS-CoV-2 multi-omics analysis [16] to further investigate their effect on SARS-CoV-2, YFV, VSV, HSV-1, and RVFV replication. Our analysis highlighted proteins that had pan-antiviral as well as virus-specific effects. The knockouts of syntaxin proteins showed virus-specific effects (Figure 1, Figure 2 and Appendix A). This was particularly evident for STX16, which displayed a strong antiviral effect for SARS-CoV-2 and was proviral for VSV. The effect on SARS-CoV-2 could be partly explained by the different roles of STX16, as this protein is necessary for late events of endocytosis, more specifically in lysosome biogenesis [41]. One additional role that could be impeded by STX16 knockout is the uptake of viral particles, as STX16 also plays an important role in vesicle formation for endocytosis [42]. SARS-CoV-2 has two main entry pathways: membrane fusion, which requires TMPRSS2 expression, or the endocytic pathway, which requires ADAM17/Cathepsin L [4] and thus the endolysosomal compartment. Lung cells, including the A549 cell line, display a negligible amount of TMPRSS2 (https://www.proteinatlas.org/, accessed on the 1 May 2023). As SARS-CoV-2 depends on endocytic uptake in this cellular model, the observed antiviral effect of STX16 might be due to the reduced entry of SARS-CoV-2 particles. In conclusion, the role of STX16 in SARS-CoV-2 biology could be more relevant for the virus uptake than for the secretion of newly generated virions, which is underlined by the very slow infection rate in STX16-depleted cells (Figure 2A and Appendix A). Conversely, VSV infection appeared to be increasing, potentially because STX16 depletion reduces the degradation of the virus through the role of STX16 in autophagosome/lysosome formation. 

GPR39, NECTIN3, and MDM2 knockout cells showed an exclusive and strong pro-viral effect for SARS-CoV-2 (Figure 1 and Figure 2). Among them, MDM2 knockout was particularly interesting as it displayed a similar phenotype to LY6E knockout cells [34]. This observation is unexpected, as MDM2 is mainly described as an E3 ubiquitin ligase involved in p53 regulation, the latter being an antiviral protein for most viruses [43,44]. Interestingly, an antiviral effect of MDM2 could be confirmed for YFV, HSV-1, and RVFV but not for SARS-CoV-2 (Appendix A), suggesting a potentially different role of MDM2 in the case of SARS-CoV-2 infection. LY6E has been shown to specifically impair SARS-CoV-2 entry, and—given the similarity of the infection phenotype in our live-cell imaging screen (Appendix A)—we speculated that MDM2 might be involved in virus uptake. This theory was substantiated by viral particle uptake assays, as SARS-CoV-2 spike-pseudotyped VSV (VSV-S) replication was significantly higher in the MDM2 knockout cells as compared to the NTC cells (Figure 2B,C).

Our combined results suggested that the increase in SARS-CoV-2 replication was an uptake-dependent effect. Knowing the enzymatic activity of MDM2 and the observed modulation of ACE2 levels in our LC–MS dataset (Figure 2D), we further investigated the contribution of the ACE2 ubiquitination site used by MDM2 derived from patient data [39]. We generated stable A549-H2B-mRFP-ACE2 and A549-H2B-mRFP-ACE2 (K788R) cell lines, which were used to elucidate the effect on SARS-CoV-2 infection and uptake. Indeed, compared to the wild-type, the K788R substitution increased the amount of ACE2, even though the transcriptional levels of ACE2 remained similar (Figure 3A,B), highlighting the role of the MDM2 ubiquitination site for ACE2 turnover. Accordingly, the SARS-CoV-2 infection was significantly increased in the ACE2 mutated cells, likely due to the accumulation of ACE2 on the cell surface and consequently increased uptake of SARS-CoV-2 (Figure 3). To further substantiate the uptake-dependent effect, we used a SARS-CoV-2 VLP system. We could clearly observe more attached particles on A549-H2B-mRFP-ACE2 (K788R) cells after 1 h of incubation as compared to A549-H2B-mRFP-ACE2 (Figure 4D). Similar results were obtained by RT-qPCR on SARS-CoV-2-infected cells (Figure 4A). Additionally, our VSV-S assay underlined the importance of the K788R substitution for SARS-CoV-2 entry, as the effect was only observed for VSV-S but not with the parental VSV strain (Figure 4B,C). The observed pro-viral effect of MDM2 knockout seems to be dependent on ACE2-mediated virus uptake (Figure 2). This is further supported as the pro-viral effect of the MDM2 knockout was SARS-CoV-2-specific (Figure 1 and Figure 2, Appendix A), which we could recapitulate by mutating K788 within ACE2 (Figure 3 and Figure 4).

Aside from MDM2, another proto-oncogene ubiquitin ligase (SKP2) has also been identified to modulate ACE2 levels. SKP2 is activated by CSE (cigarette smoke extract) and has been reported to downregulate ACE2 [45]. MDM2 appears to be more expressed in lung cells as compared to Skp2 (https://www.proteinatlas.org, accessed on 10 July 2023), even though the effect of SKP2 on ACE2 levels in lung cells was significant [45]. The coexistence of multiple ubiquitin ligases, such as MDM2 and SKP2, that can exert regulatory control over ACE2, highlights the intricate nature of ACE2 modulation. It is plausible that different ubiquitin ligases operate on ACE2 in a cell type-specific manner, depending on their expression level or specific triggers.

Our in vitro results on the function of MDM2 for SARS-CoV-2 infection could help to understand risk factors that affect disease severity. The lethality of COVID-19 can be correlated with risk factors such as pulmonary arterial hypertension and diabetes [46,47]. Recently, it has been shown that MDM2 can mediate the ubiquitination of ACE2 at position K788 and that MDM2 regulation is disrupted in patients suffering from pulmonary arterial hypertension [39]. Collectively, these data highlight the possibility that dysregulated MDM2 activity could be a risk factor for SARS-CoV-2 infection and COVID-19 pathogenicity.

MDM2 is also significantly downregulated in patients with common diabetic nephropathy [47]. It is known that SARS-CoV-2 infects the kidney, which can lead to severe kidney damage with a high mortality rate [48], and that patients suffering from diabetes are more likely to develop severe forms of COVID-19 [48,49]. Combining the latter reports with our results, one could speculate that the downregulation of MDM2 could lead to an upregulation of ACE2, subsequently leading to a higher SARS-CoV infection rate, higher tissue damage in the kidney, and an overall poor prognosis in cases of severe COVID-19. 

As ACE2 is the central receptor for SARS-CoV-2 entry, insights gained on its regulation could be a promising drug development target for treating COVID-19. Multiple small molecules have been developed to target proteasomal degradation mediated by MDM2 [50]. Interestingly, inhibition of the proteasome has been shown to stabilize ACE2, implying a potential increase in SARS-CoV-2 uptake [39]. However, despite this effect, the overall impact of proteasome inhibition on SARS-CoV-2 replication has been demonstrated to be antiviral [16,40]. This paradox may arise from the necessity of proteasome activities to promote virus replication or from the induction of endoplasmic reticulum (ER) stress, which subsequently suppresses the biosynthesis of SARS-CoV-2 proteins [40,51]. Interestingly, a proteomics study has identified over 98 ubiquitin ligases and 29 deubiquitinating enzymes that interact with SARS-CoV-2 proteins [52], underlining the relevance of this post-translational modification not just for the host but also viral proteins. Understanding this interface might lead to the development of novel therapeutic approaches, such as targeting deubiquitinating enzymes to suppress SARS-CoV-2 replication [53]. These combined findings underline the need for further research to comprehend the intricate network of the ubiquitination and deubiquitination processes. This is particularly relevant for ACE2 and its multifaceted implications in diverse physiological and pathological contexts, as it is the central receptor for SARS-CoV-2 infection.

## 5. Conclusions

We observed a SARS-CoV-2-specific pro-viral effect by MDM2 depletion, which is not evident for other viruses tested, including VSV, YFV, RVFV, and HSV-1. Proteomics analysis highlighted that the SARS-CoV-2 entry receptor ACE2 was significantly upregulated in MDM2 knockout cells, which explains the enhanced SARS-CoV-2 infectivity. The effect of the MDM2 ubiquitination site on ACE2 was further investigated by mutating the ubiquitination site K788 of the ACE2 receptor. Our results highlighted the increased protein levels of ACE2-K788R, leading to more SARS-CoV-2 infections due to a higher viral uptake. Moreover, our data suggest that the expression levels of ACE2 affect SARS-CoV-2 infectivity and that the regulation of ACE2 by cellular proteins or by potential drug treatment could be a valuable means to regulate SARS-CoV-2 infectivity.

## Figures and Tables

**Figure 1 viruses-15-01763-f001:**
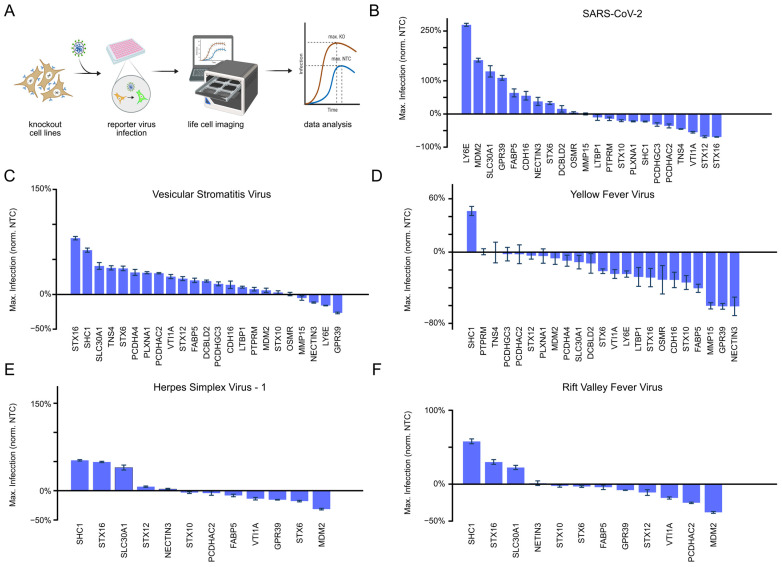
Loss of function screen identifies MDM2 as a SARS-CoV-2 restriction factor. (**A**) Experimental design and analysis of a loss-of-function screen. A549-ACE2-H2B-mRFP knockout and control cells were infected with 5 different GFP-expressing reporter viruses, and virus growth was measured using a live-cell imaging platform. The virus infection was quantified over time by normalizing the virus signal (integrated intensity of GFP) by A549-ACE2-H2B-mRFP (integrated intensity of mRFP). The maximal value during infection of infected knockout cells and NTC was used for further analysis and displayed as a bar plot in. The illustration was designed with BioRender (https://www.biorender.com/, accessed on 1 May 2023) (**B**–**F**). (**B**–**F**) Growth of SARS-CoV-2 (MOI 3), Vesicular Stomatitis Virus (MOI 0.01), Yellow Fever Virus (MOI 0.1), Herpes Simplex Virus 1 (MOI 0.1), and Rift Valley Fever Virus (MOI 0.1). Analysis was performed as described in (**A**). Each bar represents the mean (n = 3 biological repeats, +/− SEM) normalized to the NTC to show the effect of the individual gene deletion on the virus (%) compared to the infection behavior in NTC cells.

**Figure 2 viruses-15-01763-f002:**
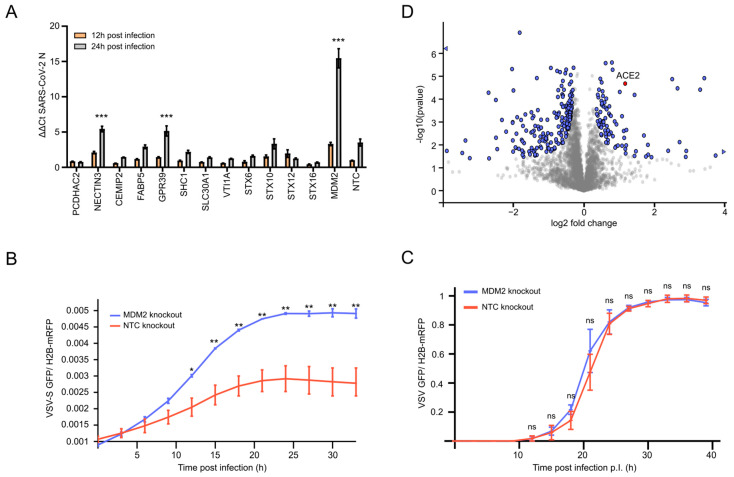
MDM2 knockout increases SARS-CoV-2 RNA abundance and supports virus uptake. (**A**) SARS-CoV-2 nucleocapsid (N) RNA levels in A549-ACE2-H2B-mRFP knockout and NTC cells infected with SARS-CoV-2 (wt) were analyzed 12 h and 24 h post-infection. SARS-CoV-2 transcripts were normalized to GAPDH and to the NTC sample. The graph shows a fold change in N abundance vs. 12 h NTC infected cells. Significance of the RNA level changes was analyzed by one-way ANOVA, comparing the 24 h levels of the knockouts to the 24 h values of the NTC, with *p* < 0.001 (***). Bars show mean signal +/− SD of 3 independent experiments. Significance was displayed when *p* < 0.001 and the difference in ∆∆Ct (12 to 24 h) of the knockout was greater than the difference in ∆∆Ct of the NTC. (**B**) A549-H2B-mRFP-ACE2 MDM2 knockout and NTC cells were infected with VSV-S-GFP reporter virus (MOI 3), and the GFP expression was measured over time (mean +/− SD, n = 3). Integrated GFP intensity was normalized to the H2B-mRFP intensity. Significance was calculated by a two-tailed Student’s *t*-test using a two-sample *t*-test with *p* < 0.05 (*), < 0.01 (**) and non-significant *p* > 0.5 (ns). (**C**) As in (**B**) but with VSV-GFP (MOI 0.01). (**D**) Full proteome analysis of A549-H2B-mRFP-ACE2 MDM2 knockout cells and NTC. Volcano plot shows 3767 analyzed proteins (dots), of which 267 proteins showed significantly different expression in MDM2 knockout vs. NTC cells (blue symbols, red for ACE2). Triangles represent proteins outside the displayed *x*-axis scaling. Significance was calculated by a two-tailed Student’s *t*-test (permutation-based FDR < 0.05, n = 4 biological replicates).

**Figure 3 viruses-15-01763-f003:**
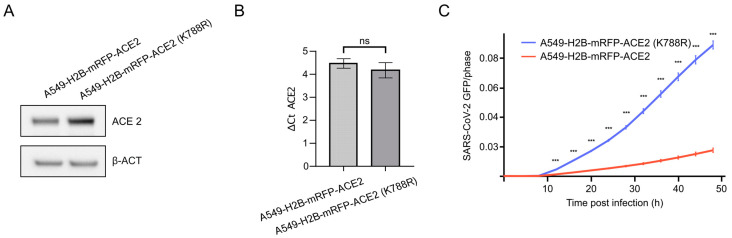
Mutation of ACE2 Lysine 788 increases protein levels of ACE2 and enhances SARS-CoV-2 infection. (**A**) Western blot showing ACE2 and β-actin (β-ACT) levels of A549-H2B-mRFP-ACE2 and A549-H2B-mRFP-ACE2 (K788R) cells. (**B**) Comparison of ACE2 mRNA levels (normalized to RPLP0) between A549-H2B-mRFP-ACE2 and A549-H2B-mRFP-ACE2 (K788R) cells. To compare mRNA levels, an unpaired two-tailed *t*-test was applied with ns: *p* > 0.05 (mean +/− SD, n = 4). (**C**) Infection of A549-H2B-mRFP-ACE2-wt and A549-H2B-mRFP-ACE2-K788R with SARS-CoV-2-GFP reporter virus. The integrated GFP area (virus signal) was normalized to the area of cells. Significance was calculated using a two-sample *t*-test, *p* < 0.001 (***).

**Figure 4 viruses-15-01763-f004:**
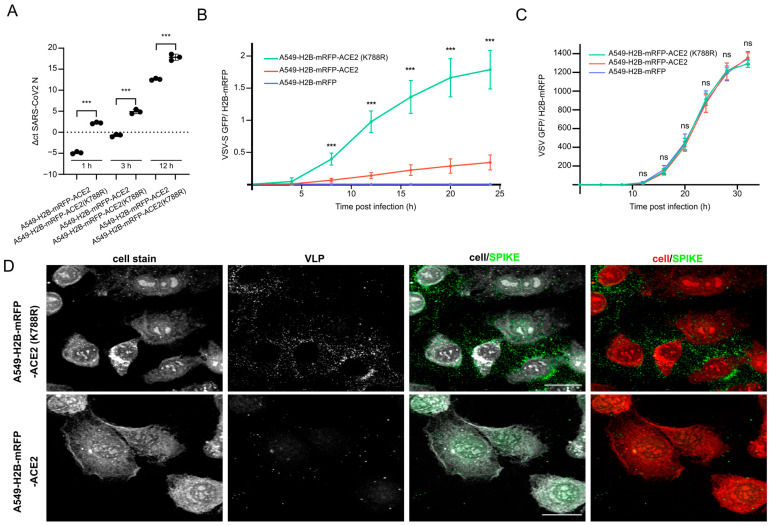
ACE2-K788R increases SARS-CoV-2 particle uptake. (**A**) Comparison of SARS-CoV-2 nucleocapsid (N) RNA levels between SARS-CoV-2 (wt) infected (MOI 3) A549-H2B-mRFP-ACE2 and A549-H2B-mRFP-ACE2 (K788R) cells at 1, 3 and 12 h post-infection. To compare the RNA levels (normalized to RPLP0 mRNA levels), an unpaired *t*-test was applied with *p* < 0.0001 (***) (mean +/− SD, n = 3). (**B**) A549-H2B-mRFP, A549-H2B-mRFP-ACE2, A549-H2B-mRFP-ACE2 (K788R), and A549-H2B-mRFP were infected with VSV-S-GFP (MOI 3), and GFP expression was measured over time (mean +/− SD, n = 3). The graph shows the ratio of the integrated intensity of the GFP signal over the RFP signal. Significance was calculated using a two-sample *t*-test with *p* < 0.001 (***). (**C**) As (**A**), but cells were infected with VSV-GFP (MOI 0.01) (mean +/− SD, n = 3). Significance was calculated using a two-sample *t*-test with *p* > 0.05 (ns). (**D**) A549-H2B-mRFP-ACE2 and A549-H2B-mRFP-ACE2 (K788R) cells were exposed to SARS-CoV-2 Spike-coated virus-like particles (VLPs). Representative immunofluorescence staining of S after VLP incubation (1 h, 37 °C). The staining was performed with the cell stain CMRA orange and a polyclonal antibody against SARS-CoV-2 Spike (scale bar, 20 µm).

## Data Availability

The files of the proteomic datasets and Maxquant output have been deposited to the ProteomeXchange Consortium (http://proteomecentral.proteomexchange.org, accessed on 1 May 2023) via the PRIDE partner repository under the project ID: PXD040891.

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
