# Peer review of "MDM2 Influences ACE2 Stability and SARS-CoV-2 Uptake"

_viruses, 2023, doi:10.3390/v15081763_

Round 1

Reviewer 1 Report

To Author:

Ubiquitination is a very important type of protein modification, which affects the normal function and stability of proteins through the protease degradation pathway. ACE2 is an important receptor of SARS-CoV-2, and there are relatively few studies on the ubiquitination of ACE2. In this study, the authors demonstrated that the E3 ubiquitin ligase MDM2 can regulate the stability of ACE2, thereby indirectly affecting the infection efficiency of SARS-CoV-2. I considered this research article to be significant. However, I have several suggestions before it can be accepted.

 Comments:

(1) In this study, the authors stated that MDM2 regulates the ubiquitination of ACE2, but there is a lack of detection of ACE2 ubiquitination. Moreover, the kind of ubiquitination on ACE2 that MDM2 regulates has not been determined.

(2) In this study, the authors proved that the ability of SARS-CoV-2 to infect cells increased after knocking out MDM2, but these were all in vitro cell experiments and further verification using animal experiments is lacking.

(3) There is one study showing that the E3 ubiquitin ligase Skp2 can also regulate the ubiquitination of ACE2 (Guizhen Wang et al. Frontiers of Medicine. 2021). The authors should compare the roles of Skp2 and MDM2 in regulating the ubiquitination of ACE2.

 (4) This manuscript contains some grammatical errors (such as line 24) and inconsistent font formatting (lines 360-362).

Reviewer 2 Report

In this manuscript Emslander and colleagues identified the ubiquitin E3-ligase MDM2 as a cellular factor which influences the ACE2 stability and thus regulates the infection rate of SARS-CoV-2.  Thereby, the authors employed an impressive range of various experimental methods.  However, there are major concerns existing:  whereas the first part of the results including the identification of MDM2 as a SARS-CoV-2 restriction factor by CRISPR/Cas-9 based knockout screening systems is well described and clear, the second part, including the biological consequences of ubiquitinylation of ACE2 at lysine 788 and subsequent proteasomal degradation remains unclear and can at the current state not be validated for publication. 

The authors should also consider of whether it would make sense to split the current manuscript into two manuscripts.  In the first one, the identification of MDM2 could be described.  In a second manuscript, the biological consequences of ubiquitinylation, deubiquitinylation as well as proteasomal degradation of ACE2 on the virus replication of SARS-CoV-2 should be analyzed.  For this second part, however, multiple studies, e.g. the usage of different replication competent virus systems and the influence of proteasome inhibitors and DUB inhibitors, as well, are absolutely necessary to underline the conclusions drawn by the authors. The manuscript as it stands requires major revisions.

Major points

1.  In the current manuscript the authors only used pseudotyped virus particles to analyze the impact of Lysine 788 of ACE2 on the replication of SARS-CoV-2.  To obtain meaningful results for the importance of K788 on SARS-CoV-2 replication the authors should perform replication studies in Calu-3 human lung cells with a "real" virus and investigate the effect of the K788R mutation on the spread of infection for example by qRT-PCR.

2. If the authors' conclusion that proteasomal degradation of ACE-2 following polyubiquitinylation of ACE-2 by MDM2 at lysine 788 decreases virus uptake would be true in virus replication, then the use of proteasome inhibitors should increase replication of SARS-CoV-2.  Therefore, the authors should perform such studies using proteasome inhibitors to confirm their statement.

3. Deubiquitinylating enzymes might counteract and prevent polyubiquitinylation of ACE2 at lysine 788, thereby increasing virus uptake.  Thus, DUB inhibitors should lead to increased polyubiquitinylation of ACE2 and thus inhibit replication of SARS-CoV-2.  However, DUB inhibitors should have no effect on the replication of a K788R mutant.  This should be investigated by the authors, especially because it has already been published that certain DUB inhibitors block SARS-CoV-2 replication.  This should additionally be discussed by the authors in the context of their own results.

4. The authors wrote the entire results part as one block.  For a better understanding for the reader, sub-headlines should be introduced in the results section.

Minor

1. On page 10, line 407-409 the authors wrote: „Importantly, this ACE2-K788 has already been identified to be the essential E3-ligase for patients suffering from idiopathic pulmonary arterial hypertension.“. It´s not clear what the authors mean with this sentence. The ACE-K788 is not an E3-ligase. In the same paragraph, the authors continue: „It has been shown that MDM2 is upregulated in these patients and that MDM2 ubiquitinates ACE2 at position K788, which leads to a subsequent downregulation of ACE2 through proteasomal degradation (Shen et al. 2020).“ If the authors' conclusions were correct, then these patients would be less likely to contract SARS-CoV-2. Is there any epidemic evidence for this? The authors should at least discuss this point.

2. Some parts of the figure legend, e.g. figure legend 1 or 2 have a different font. This should be corrected by the authors.

3. In Figure 3 A the authors show Western blot results and conclude from them that the ACE2-K788R mutant contain higher intracellular levels of ACE2 as the wt. This conclusion is not convincingly evident from the Western blot shown. Thus, the authors should also perform a densitometric evaluation of these experiments.

Round 2

Reviewer 2 Report

As before, the manuscript requires major revisions, including replications studies.  Although now discussed in the context of the literature, sill it would be important to investigate how proteasome and DUB inhibitors replication of both, wt and K788.  Given the fact, that none of my major previous requests have been fulfilled, the ms is not acceptable for publication. 
